# Yellow Fever: Roles of Animal Models and Arthropod Vector Studies in Understanding Epidemic Emergence

**DOI:** 10.3390/microorganisms10081578

**Published:** 2022-08-05

**Authors:** Divya P. Shinde, Jessica A. Plante, Kenneth S. Plante, Scott C. Weaver

**Affiliations:** 1World Reference Center for Emerging Viruses and Arboviruses, University of Texas Medical Branch, Galveston, TX 77555, USA; 2Department of Microbiology and Immunology, University of Texas Medical Branch, Galveston, TX 77555, USA; 3Center for Vector-Borne and Zoonotic Diseases, University of Texas Medical Branch, Galveston, TX 77555, USA; 4Institute for Human Infections and Immunity, University of Texas Medical Branch, Galveston, TX 77555, USA

**Keywords:** yellow fever virus, animal models, arthropod vectors

## Abstract

Yellow fever virus (YFV) is a mosquito-borne flavivirus circulating throughout the tropical and sub-tropical regions of Africa and South America. It is responsible for an estimated 30,000 deaths annually, and while there is a highly successful vaccine, coverage is incomplete, and there is no approved treatment for YFV infection. Despite advancements in the field, animal models for YFV infection remain scarce, and care must be taken to select an appropriate model for a given hypothesis. Small animal models require either adapted YFV strains or immunocompromised hosts. Non-human primates (NHPs) recapitulate human disease, but they require specialized facilities and training, are often in short supply and cost-prohibitive, and can present ethical concerns. The limitations in studying the mosquito vectors for YFV infection include inconsistency in the laboratory environment, the requirement for a high containment insectary, and difficulty in maintaining sylvatic mosquitoes. In this review, we discuss the roles of animal models and arthropod vector studies in understanding epidemic emergence.

## 1. Introduction

Yellow fever virus (YFV) is a mosquito-borne virus of the genus Flavivirus and family Flaviviridae [1]. It has a positive sense, single-stranded RNA genome approximately 11 kb long [2]. YFV primarily circulates in three cycles: urban, sylvatic (enzootic), and intermediate (savannah) (Figure 1) [3]. In the urban cycle, YFV is transmitted between peridomestic mosquito species chiefly, *Aedes aegypti* and humans who serve as amplification hosts [4]. In the sylvatic cycle, YFV is transmitted between non-human primates (NHPs) and sylvatic mosquitoes including *Ae. africanus* in Africa [5] and Haemogogus spp. and Sabethes spp. in South America [3,6]. The enzootic vectors in South America can occasionally infect humans, an occurrence that resulted in a massive outbreak from 2016–2020. The intermediate cycle exists only in Africa in rural areas that border forest or savannah and involves the transmission of the YFV between both humans and NHP hosts and semi-domestic mosquito vectors such as *Ae. furcifer, Ae. bromeliae, Ae. luteocephalis* etc. [5]. The intermediate transmission can occur in areas with some human activity such as village settlements where humans can come in contact with infected semi-domestic mosquitoes [7].

Despite the availability of a safe and effective vaccine for nearly a century, yellow fever (YF) disease affects approximately 200,000 individuals, causing an estimated 30,000 deaths annually [8]. YFV continues to cause periodic, large outbreaks in Africa [4] and South America [9]. During 2016, Angola recorded 4307 suspected cases and 306 suspected deaths [4], and Brazil recorded 2251 cases and 772 deaths [9]. This ongoing high level of circulation combined with recent vaccine shortages in the face of outbreaks [10] raises alarms over the risk of importation to areas with immunologically naïve populations such as Asia and North America. Moreover, vaccinations in these areas are not widespread due to difficulty in manufacturing, and implementing it would further increase the strain on vaccine supplies. Additionally, there are restrictions with vaccinating everyone due to complications associated with adverse events, especially in the elderly [11].

YFV infection causes varying levels of disease characterized by asymptomatic to mild flu-like illness ranging to hemorrhagic manifestations and death [12]. Severe YF is a systemic illness characterized by high viremia, hepatic, renal, and myocardial injury, and hemorrhage with a case fatality rate of 20–50% [12]. In humans, the incubation period is typically 3–6 days [13], and the disease develops in three stages (Figure 2). The “period of infection” is the viremic phase with non-specific signs and symptoms such as malaise, headache, nausea, and fever. This is followed by the “period of remission”, wherein the symptoms remit in most cases, and patients recover. However, one in seven persons progress to the “period of intoxication” where they develop the severe viscerotropic disease with an enlarged and tender liver, renal dysfunction, jaundice, cardiovascular instability, and hemorrhage, which can lead to death [12].

## 2. Brief History of YFV Research

YFV most likely originated in Africa and arrived in the Americas with the slave trade in the 1600s [14]. Parasitologist Patrick Manson, who studied filariasis, was the first to suggest the role of mosquitoes as intermediate hosts of a pathogen [15]. His investigations influenced the work of other scientists to discover the mode of transmission of yellow fever. The most significant findings were by Carlos Juan Finlay, a Cuban physician who showed a local mosquito to be a probable vector of YFV and studied mosquito structure, biology, and behavior [16]. In 1881, Finlay hypothesized that the YFV was transmitted by *Aedes aegypti* (previously called *Culex fasciatus*) mosquitoes [17], representing the first time that an arthropod vector had been proposed for any virus. Finlay’s theories were confirmed by United States Army pathologist Walter Reed and collaborators, who proved that the virus was arthropod-borne, and that mosquitoes were the putative vector [18]. The discovery of the mode of transmission quickly led to measures by the American surgeon William Crawford Gorgas to eradicate the vector in Havana, Cuba. These actions lead to a precipitous decline in yellow fever cases [19]. The successful elimination of large YFV outbreaks as a result of vector control efforts was noted and soon replicated by other countries, including Brazil and Panama. One of the greatest challenges during the construction of the Panama Canal was massive deaths occurring due to mosquito-borne diseases such as yellow fever and malaria. Successful vector control efforts in this area facilitated the completion of the Panama Canal [19].

In 1927, British physician Adrian Stokes isolated YFV from a Ghanian patient, Mr. Asibi, for whom this prototypical YFV strain is named [20]. This groundbreaking work was the first time that a human virus had been isolated. Unfortunately, Stokes contracted the virus during his experiments and died within four days [20]. In 1937, the live-attenuated 17D vaccine was obtained by passaging the Asibi strain of YFV a total of 176 times, initially in live monkeys and murine embryonic tissues, followed by chicken embryos and eventually chicken embryos lacking neurological tissue; this work earned Max Theiler the 1951 Nobel Prize in medicine [21].

Due to the devastating outbreaks, in 1942 the Pan American Health Organization (PAHO) started an ambitious vector-control program with the goal of completely eradicating *Ae. aegypti* mosquitoes utilizing insecticides such as DDT and source reduction to remove artificial containers that usually serve as larval habitats [22,23]. By 1962, *Ae. aegypti* was eradicated in almost 20 Latin American countries, including Brazil [22,24]. However, urbanization, transportation, insecticide resistance, concerns over off-target effects of DDT, lack of funds and political will resulted in the reinfestation of these mosquitoes throughout the Americas [25].

## 3. Animal Models

### 3.1. Mice (Mus Musculus)

Laboratory rodents display varying degrees of susceptibility to YF with multiple factors playing important roles such as the virus strain, animal age and immune status, and route of and dose of infection. While model development with wild-type strains of YFV has been limited, partially due to biosafety requirements, numerous efforts have been made in developing a mouse model using the vaccine strain 17D and its sub-strains. Table 1 summarizes the attempts for mouse model development using wild-type (WT) and vaccine strains at different doses and routes of inoculation in immunocompetent and immunocompromised mice. Efforts to generate a lethal, viscerotropic model in immunocompetent mice have thus far proven unsuccessful. In contrast, several models exist using mice that lack key component(s) of the innate immune response. Mice lacking both the IFN-α/β receptor and the IFN-γ receptor are susceptible to the 17D-204 vaccine, developing a lethal infection in a dose-dependent fashion with high viral titers in the brain and liver [26]. Three- to four-week-old mice lacking either the IFN-α/β receptor (A129) or the STAT1 signaling molecule (STAT129) are susceptible to wild-type strains of YFV such as Asibi and Angola73, succumbing to illness within six to seven days [27]. Elevated MCP-1 and IL-6 levels in these mice suggest the development of a cytokine storm, and histopathological examination of the liver revealed extensive damage including fatty acid steatosis, apoptotic and necrotic lesions, and immune cell infiltrate [27]. All these factors are consistent with the viscerotropic, hepatic YF disease seen in severe human illness. Similar observations are reported in adult C57BL/6 IFNAR^−/−^ mice (lacking type 1 interferon receptor on C57BL/6 background) [28]. Studies in STAT2 ^−/−^ (lacking signal transducer and activator of transcription 2) mice have shown efficient replication of YFV Asibi stain emphasizing the role of murine STAT2 in restricting replication [29]. In contrast to observations in other flaviviruses such as DENV and ZIKV. hSTAT2 incorporated in murine cells does not bind to YFV NS5, indicating the role of other human factors [29]. Additional examples of the disease outcome in immunocompromised mice are summarized in Table 1. Thus, immunocompromised mouse models are a useful tool for examining YFV pathogenesis and testing vaccine and therapeutic candidates, provided that the early, innate immune response is not critical to the scientific question under examination.

Another strategy for the development of a YF mouse model has been chimerization. The FRG (fumarylacetoacetate) mice have three genetic lesions (Fah^−/−^, Rag^−/−^, Il2rg^−/−^) on a C57BL/6 background. They are immune-deficient Fah knockout mice, and they lack the genes for Rag-2 and the common gamma chain of the interleukin receptor [30]. These mice can then be modified to replace murine with human hepatocytes (hFRG) [31]. Intravenous inoculation of hFRG mice with 2 × 10^5^ focus-forming units (ffu) of a highly pathogenic YFV-Dakar strain resulted in 10–25% weight loss and a viscerotropic disease in conjunction with high viral titers in the liver [31]. Histopathological analysis of YFV-infected hFRG livers showed apoptotic and steatotic hepatocytes, areas of necrosis, and inflammatory infiltration, all of which are classic YF manifestations in humans [31]. Despite the shortcomings of current YF mouse models, all of which require either dysregulation of the animals’ immune response or chimerization, they represent an important resource due to the cost-effectiveness, availability of both animals and reagents, and simplicity of working with mice in contrast to hamsters or NHPs. Efforts toward the development of an immune-intact, adult mouse model capable of mimicking human illness would be of substantial benefit to YFV research.

**Table 1 microorganisms-10-01578-t001:** Mouse models of YFV infection.

Mouse	Age	Virus Strain	Route of Inoculation/Dose	Disease Outcome	Reference
Immunocompetent					
C57BL/6	3–4-week-old	17D *	^1^ IM; 10^4^, 10^7 4^ PFU	No signs of disease up to 30 days	[32]
			^2^ IP; 10^4^ PFU	No signs of disease up to 30 days	[32]
WT129	3–4-week-old	17D-204 ^#^	^3^ SQ (both footpad); 10^6^ PFU	No signs of disease	[27]
		Asibi; ^#^ Angola73 ^#^	SQ (both footpad); 10^6^ PFU	No signs of disease	[27]
Immunocompromised					
C57BL/6 (IFNAR^−/−^)	3–4-week-old	17D *	SQ; 10^4^ PFU	No signs of disease	[32]
			Footpad; 10^4^ PFU	Transient swelling at site of inoc; fully recovered by 6 DPI	[32]
			IM; 10^4^ PFU	32% of mice had neurotropic or viscerotropic disease	[32]
			IM; 10^7^ PFU	42% of mice had neurotropic or viscerotropic	[32]
	6–7-week-old	17D *	IM; 10^7^ PFU	No signs of disease	[32]
	Independent of age	Angola71 ^#^	SQ (both rear footpads); 2 × 10^4^ PFU	100% lethal by 6–9 DPI	[28]
A129	3–4-week-old	17D-204 ^#^	SQ (both footpad); 10^6^ PFU	No signs of disease	[27]
		Asibi; ^#^ Angola 73 ^#^	SQ (both footpad); 10^4^ PFU	Clinical signs such as lethargy, hunched posture, swelling at inoculation site; weight loss; lethal by 7–8 DPI	[27]
AG129	3–4-week-old	17D-204 ^#^	SQ (both footpad); 10^6^ PFU	Weight loss; lethal by 10–11 DPI	[27]
	6–7-week-old	17D-204 ^†^	IP; 10^4^ PFU	Neurotropic; lethal; ^5^ AST (days): 17.2 ± 1.1	[26]
			IP; 10^5^ PFU	Neurotropic; lethal; AST (days): 15.2 ± 3.3	[26]
			IP; 10^6^ PFU	Neurotropic; lethal; AST (days): 12.0 ± 3.6	[26]
			SQ; 10^4^ PFU	Neurotropic; 70% mortality; AST (days): 20.8 ± 4.6	[26]
			SQ; 10^5^ PFU	Neurotropic; lethal; AST (days): 16.8 ± 2.1	[26]
	3–4-week-old	Asibi; ^#^ Angola73 ^#^	SQ (both footpad); 10^4^ PFU	Clinical signs such as lethargy, hunched posture, swelling at inoculation site; weight loss; lethal by 6–7 DPI	[27]
G129	3–4-week-old	17D-204 ^#^	SQ (both footpad); 10^6^ PFU	No signs of disease	[27]
		Asibi; ^#^ Angola73 ^#^	SQ (both footpad); 10^4^ PFU	No signs of disease	[27]
STAT129	3–4-week-old	17D-204 ^#^	SQ (both footpad); 10^6^ PFU	No signs of disease	[27]
		Asibi ^#^	SQ (both footpad); 10^4^ PFU	Lethal by 7–8 DPI	[27]
STAT2^−/−^	Unknown	Asibi	Hock; 10^4^ PFU	Viral burden and histopathological abormalities in spleen, liver; viremia on day 3 PI	[29]

^1^ IM, intramuscular; ^2^ IP, intraperitoneal; ^3^ SQ, subcutaneous; ^4^ PFU, plaque-forming unit; ^5^ AST, average survival time. * Virus infectivity determined in Baby Hamster Kidney (BHK) cells; ^#^ Virus infectivity determined in human hepatoma derived cells (Huh7); ^†^ Virus infectivity determined in African green monkey epithelial cells (Vero E6).

### 3.2. Hamsters (Mesocricetus Auratus)

*Mesocricetus auratus*, commonly known as the Syrian golden hamster, offers an alternative small animal model for the study of YF. In contrast to mouse models, which rely on modification of the host but permit infection with wild-type strains of YFV, the hamster models utilize wild-type animals with hamster-adapted strains of YFV. The Jiménez strain of YFV was hamster adapted by ten passages in hamster livers, and the resulting Jiménez p10 strain induced lethargy, loss of appetite, and ruffled fur, as well as generated a higher viremia than the parental p0 Jimenez strain [33]. Histopathologic analyses showed acidophil bodies compatible with the apoptotic bodies frequently observed in the livers of experimentally infected rhesus macaques and fatal human cases [34]. Other findings in the livers of YFV Jimenez-infected hamsters were massive hepatocyte necrosis in conjunction with elevated levels of serum aspartate and alanine amino transaminase and microvesicular steatosis, all of which are consistent with human disease [34]. Molecular characterization of the Jimenez p10 strain revealed five nucleotide changes encoding two amino acid substitutions within the non-structural proteins NS3 and NS5 [35]. A hamster-adapted Asibi strain of YFV was also developed by seven passages in hamster livers, resulting in Asibi/hamster p7 virus that caused robust viremia, severe illness, viscerotropic disease, and death in subadult hamsters by 8 days post-infection (DPI) [36]. In comparison to the parental Asibi strain, the genome of the Asibi/hamster p7 virus had 14 nucleotide changes encoding seven amino acid substitutions (five of which were in the envelope protein) [36]. The above findings suggest that there are multiple molecular mechanisms responsible for the viscerotropism in hamsters. Nevertheless, the hamster model can be utilized for several applications, especially for vector competence experiments that require high levels of viremia [37].

The hamster model has been used extensively for the evaluation of antiviral drugs and vaccine candidates, applications for which the intact immune system of the hamster model is particularly important [38,39,40,41]. T-705, a small molecule therapeutic shown to be active against influenza virus, and T-1106, a chemical analog of T-705, were effective in the treatment of YFV Jimenez-infected hamsters with improved survival, reduced viremia, and reduced serum levels of liver enzymes [38,39]. The early treatment of YFV Jimenez-infected hamsters with ribavirin also resulted in reduced liver damage and increased survival [40]. The YFV Jimenez hamster model was also used to determine the efficacy and immune correlates of protection of XRX-001, an inactivated whole virion YFV vaccine candidate adjuvanted with aluminum hydroxide [41,42].

### 3.3. Non-Human Primates

Historically, NHPs have been used as models to recapitulate human disease. In addition to serving as models of human infection, NHP infection with YFV is worthy of study in and of itself due to its role as a natural reservoir in sylvatic and intermediate transmission cycles. Neotropical NHPs such as *Alouotta* spp. (howler monkeys), *Callithrix* spp. (marmoset) and *Callicebus* spp. (titis) are susceptible to YFV infection [43], and fatal epizootics often precede human infections [9]. Despite the efforts in surveillance and vaccinations, the 2016–2019 YFV outbreak in Brazil is considered the most significant New World YF activity in the last 70 years, resulting in 2251 human cases and 772 deaths as well as 2590 confirmed NHP infections [9,43]. Alternatively, African NHPs such as the *Cercopithecus* spp. (genons)., *Cercocebus* spp. (mangabey), *Colobus* spp. (colobus) are susceptible to YFV infection but are mostly resistant to disease making, it difficult to isolate the virus and use these NHPs as sentinels [44,45]. Various attempts at experimental infection of NHPs are summarized in Table 2.

The first documented experiments with NHPs and YFV date to 1928 when Stokes and colleagues discovered that monkeys from Asia (rhesus macaques) were susceptible to infection with almost identical clinical and post-mortem reports as those from humans [46]. Thereafter, rhesus and cynomolgus macaques have commonly been used to study the clinical, biochemical, immunological, and pathophysiologic aspects of YF [47,48]. Histological observations in rhesus macaques (infected with doses ranging from 25 to 5 × 10^4^ TCID_50_ of YFV strain DakH1279) showed a classic viscerotropic disease with significant injury to the liver and kidney [47]. Infected livers were discolored with signs of hemorrhagic foci, extensive hepatocyte necrosis, and councilman bodies (eosinophilic degeneration of liver cells), and all the kidney sections showed evidence of tubular necrosis and protein deposits. Consistent with these gross pathological findings, key indicators of liver and kidney damage such as lymphopenia, elevated cytokines, and elevated liver enzymes were often observed only a few hours before humane euthanasia of moribund animals [47,48].

Experimental infection of African NHPs with YFV produces an inapparent infection and only mild disease. In contrast, many African NHPs have high, intense, and prolonged viremia, making them good amplifying hosts for the virus. Since epizootic NHP deaths associated with YFV in Africa are not as common as in South America, surveillance is of utmost importance, especially since multiple species of NHPs are often found to be seropositive [49].

YFV antibodies have been detected in chimpanzees (*Pan troglodytes*) and baboons (*Papio cynocephalus*, *Papio anubis*) during surveillance in Africa. Experimental infection of chimpanzees with an African YFV strain showed relatively low viremia that lasted for a duration of 3 days [50]. Inoculation of African pottos (*Perodicticus potto*) with YFV resulted in 4–8 days of viremia high enough to infect mosquitoes; however, no clinical signs were observed up to 10 DPI [49].

In the New World, many species of NHPs are impacted during epizootics in terms of severity of disease and fatality with the most common ones being marmosets (*Callithrix* spp.), howler monkeys (*Alouatta* spp.), night monkeys (*Aotus* spp.), spider monkeys (*Ateles* spp.), woolly monkeys (*Lagothrix* spp.), squirrel monkeys (*Saimiri* spp.), and capuchin monkeys (*Cebus* spp.), [44]. Susceptibility and tolerance to infection with YFV varies greatly among these groups and even among species of these groups. Howler monkeys are generally considered the most susceptible species to minimal doses of experimental YFV infection [51]. In contrast, the capuchin monkeys are relatively refractory to infection [52]. Animals that survive infection typically develop lifelong humoral immunity against YFV, which allows for serological analysis.

**Table 2 microorganisms-10-01578-t002:** NHP models of YFV Infection.

NHP Species	YFV Strain	Dose/Route of Inoculation	Outcome	Reference
Old World NHPs				
*Macaca mullata* (Rhesus macaque)	DakH129 *	25 ^1^ TCID_50_; ^2^ SQ	75% lethal (3/4) by 5–7 DPI; 39–79% hepatocellular damage in animals with lethal outcome	[47]
		10^2^ TCID_50_; SQ	75% lethal (3/4) by 5–7 DPI; 36–57% hepatocellular damage in animals with lethal outcome	[47]
		10^3^ TCID_50_; SQ	84% lethal (5/6) by 4–6 DPI; 33–77% hepatocellular damage in animals with lethal outcome	[47]
		10^4^ TCID_50_; SQ	50% lethal (1/2) by 4–5 DPI; 68% hepatocellular damage in the animal with lethal outcome	[47]
		5 × 10^4^ TCID_50_; SQ	75% lethal (3/4) by 4–5 DPI; 70–81% hepatocellular damage in animals with lethal outcome	[47]
	Asibi	800 adult mouse ^3^ ic LD_50_; ^4^ IP	100% lethal (3/3) by 5–6 DPI	[53]
	French Viscerotropic	8000 baby mouse ic LD_50_; ^5^ ID	100% lethal (2/2) by 5–6 DPI	[53]
	JSS (South American)	3 baby mouse ic LD_50_; SQ	0% lethal (0/2)	[53]
*Macaca fascicularis* (Cynomolgus macaque)	Asibi ^#^	900 ^6^ PFU; SQ	100% lethal (1/1) by Day 7 PI; Peak viremia 9 log_10_ PFU/mL; elevated levels of ^7^ AST and ^8^ ALT	[54]
	Asibi ^#^	10^4^ TCID_50_; SQ	100% lethal (6/6) by 6 DPI; liver swelling, jaundice, pathological abnormalities in liver, spleen, lymph node, kidney; peak viremia 3 × 10^9^ genome copies/mL; increased ALT, AST, AP, Bilirubin	[55]
	17D ^#^	10^4^ TCID_50_; SQ	No clinical signs of disease	[55]
*Galago crassicaudatus* (Bush baby)	Unknown	Unknown	50% lethal; viremia duration 4–8 days	[44]
*Cercopithecus* spp. (various species)	Unknown	Unknown	Intense viremia for 3–4 days followed by immunity	[44]
*Colobus abyssinicus*	East African strain	2–6000 mouse LD_50_	No apparent signs of disease; Viremia duration 6–9 days	[45]
*Cercocebus* spp.	Unknown	Unknown	Viremia duration 2 days (Only *C. torquatus* develops high enough viremia to infect mosquitoes)	[44]
*Pan troglodytes (Chimpanzee)*	African	400 mouse LD_50_	Viremia duration 3 days	[50]
*Perodicticus potto (African potto)*			Intense viremia, high enough to infect mosquitoes; viremia duration 4–8 days; no clinical signs of disease	[49]
New World NHPs				
*Saimiri* spp. (Squirrel monkey)	BeH655417 * (South American)	1 × 10^6^ PFU/mL; ID	Viscerotropic disease; pathological abnormalities and viral burden in liver, spleen, kidney, lymph node, heart, lung, brain, stomach	[56]
*Callithrix albicollis* (Marmoset)	Asibi	Unknown (blood transferred from other NHPs or via mosquito bites)	33% lethal (2/6); febrile, pathological abnormalities in liver and kidney	[57]
	S.R.	Unknown (blood transferred from other NHPs or via mosquito bites)	80% lethal (4/5); febrile, pathological abnormalities in liver, kidney, spleen	[57]
*Leontocebus ursulus* (Tamarin)	Asibi	Unknown (blood transferred from other NHPs or via mosquito bites)	100% lethal (5/5); febrile, pathological abnormalities in liver and kidney	[57]
*Cebus* spp. (Capuchin)	Asibi; S.R.	Unknown (blood transferred from other NHPs or via mosquito bites)	Very low mortality, viremia may or may not be high enough to infect mosquitoes	[58]

^1^ TCID_50_, tissue culture infectious dose 50; ^2^ SQ, subcutaneous; ^3^ IC LD_50_, inoculum titrated by intracerebral inoculation of mice with serial tenfold dilutions of the virus; ^4^ IP, intraperitoneal; ^5^ ID, intradermal; ^6^ PFU, plaque-forming units; ^7^ AST: aspartate aminotransferase; ^8^ ALT: alanine aminotransferase; * Virus infectivity determined in *Aedes albopictus* cell line (C6/36); ^#^ Virus infectivity determined in African green monkey epithelial cells (Vero E6).

### 3.4. Other Animal Models

Attempts to experimentally infect other animal models with YFV have been unsuccessful. Guinea pigs inoculated by the intraperitoneal route with an infectious dose of 10^6^ TCID_50_ of the YFV strains Asibi, Jimenez as well as the hamster adapted strains Asibi/hamster p7 and Jimenez/hamster p10, showed no clinical signs of illness, no detectable viremia and 100% survival [35]. Histopathological observation of liver, spleen and brain showed no abnormalities [35]. Historic experiments have shown that rabbits are non-susceptible to YF disease with no apparent clinical signs and detectable viremia, but they develop immunity [59]. Other attempts to evaluate the immune response and determine the level of circulating virus in blood in experimentally YFV-exposed wild animals displayed varied results [60]. *Akodon* spp. (South American grass mice), *Oryzomys* spp. (North American semiaquatic rodent), and *Didelphis paraguayensis* (Brazilian opossum) were completely negative with an absence of detectable viremia and neutralizing antibody response, suggesting that these animals play no role in epidemic emergence [60]. *Metachirops opossum* (South American opossum) had inconsistent and low viremia, although neutralizing antibodies were detected in the majority of the animals exposed, suggesting that they could be dead-end hosts [60]. In contrast, *Metachirus nudicaudatus* (brown four-eyed opossum) were found to be susceptible with viremia high enough to infect mosquitoes and short-term neutralizing antibody response [60]. *Trinomys dimidiatus* (soft-spined Atlantic spiny rat) and armadillos had inconsistent and low viremia and unreliable neutralizing antibodies, indicating no epidemiological significance [60].

Based on the purpose of the experiment, one animal model may be preferred over the other. The advantages and disadvantages of common laboratory animal models are listed in Table 3.

## 4. Arthropod Vectors

### 4.1. The Urban Cycle of Transmission

*Aedes aegypti*, also known as the “yellow fever mosquito”, was most likely introduced into the New World approximately 400–500 years ago from West Africa via the European slave trade [61]. Despite its limited flight range, *Ae. aegypti* has expanded its geographic range rapidly due to international trade and globalization [61]. The eggs are resistant to desiccation, which enables their survival during long-distance travel [62]. Although *Ae. aegypti* are found throughout tropical and sub-tropical regions, their populations vary in their ability to transmit arboviruses. Outside of Africa, *Ae. aegypti* populations are anthropophilic, which contributes to their vectorial capacity for human pathogens [63]. In contrast, *Ae. aegypti* populations across sub-Saharan Africa vary strongly in their ecology, behavior, and appearance [62]. Some populations are tree dwelling and are found in forests feeding on other mammals [64,65], whereas the more “domesticated” forms are found around homes feeding on humans [64,65]. *Aedes aegypti* females lay eggs in multiple, small batches, and the domestic form *Ae. aegypti aegypti* is typically found in human settlements and oviposits in man-made containers [63,64,65]. These daytime feeders are competent vectors for several medically relevant arboviruses such as dengue, chikungunya, Zika, and YFV.

Although urban, human-amplified YF outbreaks occurred frequently in the Americas for centuries, major urban outbreaks had not been documented since 1942. The 2016–2019 Brazilian outbreak was due to spillover infections from the sylvatic cycle, with no detection of human-amplified transmission [6]. However, the presence of *Ae. aegypti* in nearly all of the affected regions has raised major concerns over the specter of an urban cycle establishing itself in South America [9]. This fear was driven by various studies (summarized below and in Table 4) that have shown Brazilian *Ae. aegypti* mosquito colonies to be highly susceptible to YFV infection [66,67,68].

Vector competence is defined as the intrinsic ability of a mosquito to acquire and transmit a pathogen [69]. Vector competence studies were conducted with three Brazilian *Ae. aegypti* colonies (Manaus, Goiania, and Rio) experimentally infected with two Brazilian and one Senegalese YFV strain at an infectious dose of 10^6^ PFU via artificial bloodmeal. At 14 DPI, the three mosquito colonies exhibited a significant difference in infection rates for the three YFV strains ranging from 30–85% [66]. Vector competence studies with *Ae. aegypti* collected at eight different locations across Senegal and infected with two West African YFV strains had great variation in the midgut infection rate (11–100%) and dissemination rate (0–59%) across all locations as well as between the two YFV strains [70]. Another study to evaluate the vector competence of Australian *Ae. aegypti* from two different locations (Cairns and Townsville) showed that mosquitoes exposed to an artificial bloodmeal with a titer of 6.7 Log_10_ TCID_50_/mL of a South American YFV strain had 24% and 36% infection rates, whereas the mosquitoes exposed to 7 Log_10_ TCID_50_/mL of a Nigerian YFV strain had 80% and 72% infection rates, respectively [71]. All these results further support the variability in results and complexity in vector competence studies. A summary of various vector competence studies in *Ae. aegypti* from different locations has been listed in Table 4. Increased knowledge of vector competence and transmission of viruses will improve our ability to predict and respond to emerging arboviruses through better surveillance and vector control methods.

**Table 4 microorganisms-10-01578-t004:** Vector competence studies in *Ae. aegypti* mosquitoes.

Colony	Infectious Dose/Route	YFV Strain	Results	Reference
Santos, Brazil	7–7.8 Log_10_ ^1^ PFU/mL; ^2^ ABM	Brazilian (MG2001)	^4^ IR: 35%, ^6^ TR: 28% at 11–14 ^3^ DPF	[72]
	6.3 Log_10_ PFU/mL; ABM	Brazilian (MG2001)	TR: 23% at 21 DPF	[72]
Respublic of Vanuatu	7–7.8 Log_10_ PFU/mL; ABM	Brazilian (MG2001)	IR: 18%, TR: 12% at 11–14 DPF	[72]
	6.3 Log_10_ PFU/mL; ABM	Brazilian (MG2001)	TR: 17% at 21 DPF	[72]
Goias, Brazil	6 Log_10_ PFU/mL; ABM	Brazilian (4408-1E)	IR: 0, ~30, ~70, ~10 % at 3, 7, 14 and 21 DPF ^5^ DR: 0, ~25, ~70, 100% at 3, 7, 14 and 21 DPF TR: 0, 0, ~20, ~50% at 3, 7, 14 and 21 DPF	[66]
		Brazilian (74018-1D)	IR: 0, ~30, ~80, ~65% at 3, 7, 14 and 21 DPF DR: 0, ~35, ~65, ~90% at 3, 7, 14 and 21 DPF TR: 0, 0, ~20, 0% at 3, 7, 14 and 21 DPF	[66]
		Senegalese (S-79)	IR: 0, ~30, ~80, 0% at 3, 7, 14 and 21 DPF DR: 0, ~35, ~75,0% at 3, 7, 14 and 21 DPF TR: 0% at 3, 7, 14 and 21 DPF	[66]
Manaus, Brazil	6 Log_10_ PFU/mL; ABM	Brazilian (4408-1E)	IR: ~55%, DR:~85%, TR: ~25% at 14–21 DPF	[66]
		Brazilian (74018-1D)	IR: ~55%, DR: ~60%, TR: ~15% at 14–21 DPF	[66]
		Senegalese (S-79)	IR: ~30%, DR: ~50%, TR: ~35% at 14–21 DPF	[66]
Rio, Brazil	6 Log_10_ PFU/mL; ABM	Brazilian (4408-1E)	IR: ~85%, DR: ~60%, TR: ~60% at 14–21 DPF	[66]
		Brazilian (74018-1D)	IR: ~45%, DR: ~60%, TR: ~35% at 14–21 DPF	[66]
		Senegalese (S-79)	IR: ~50%, DR: ~65%, TR: ~40% at 14–21 DPF	[66]
Fatick, Senegal	6.22 Log_10_ PFU/mL; ABM	Nigerian (BA-55)	^7^ MIR: 100%, ^8^ DIR:59% at 14 DPF	[70]
	5.9 Log_10_ PFU/mL; ABM	Nigerian (DAK 1279)	MIR: 17%, DIR: 0% at 14 DPF	[70]
Bignona, Senegal	6.22 Log_10_ PFU/mL; ABM	Nigerian (BA-55)	MIR: 83%, DIR: 13% at 14 DPF	[70]
	7.79 Log_10_ PFU/mL; ABM	Nigerian (DAK 1279)	MIR: 33%, DIR:0% at 14 DPF	[70]
Richard Toll, Senegal	6.32 Log_10_ PFU/mL; ABM	Nigerian (BA-55)	MIR: 57%, DIR: 10% at 14 DPF	[70]
	7.79 Log_10_ PFU/mL; ABM	Nigerian (DAK 1279)	MIR: 57%, DIR: 17% at 14 DPF	[70]
Goudiry, Senegal	6.04 Log_10_ PFU/mL; ABM	Nigerian (BA-55)	MIR: 53%, DIR: 0% at 14 DPF	[70]
	5.9 Log_10_ PFU/mL; ABM	Nigerian (DAK 1279)	MIR: 10%, DIR: 0% at 14 DPF	[70]
Kedougou, Senegal	5.34 Log_10_ PFU/mL; ABM	Nigerian (BA-55)	MIR: 35%, DIR: 0% at 14 DPF	[70]
	5.9 Log_10_ PFU/mL; ABM	Nigerian (DAK 1279)	MIR: 10%, DIR: 0% at 14 DPF	[70]
PK10, Senegal	6.04 Log_10_ PFU/mL; ABM	Nigerian (BA-55)	MIR: 27%, DIR: 3% at 14 DPF	[70]
	5.9 Log_10_ PFU/mL; ABM	Nigerian (DAK 1279)	MIR: 22%, DIR: 0% at 14 DPF	[70]
Mont Rolland, Senegal	6.2 Log_10_ PFU/mL; ABM	Nigerian (BA-55)	MIR: 27%, DIR: 0% at 14 DPF	[70]
	5.9 Log_10_ PFU/mL; ABM	Nigerian (DAK 1279)	MIR: 20%, DIR: 3% at 14 DPF	[70]
Rufisque, Senegal	6.13 Log_10_ PFU/mL; ABM	Nigerian (BA-55)	MIR: 17%, DIR: 0% at 14 DPF	[70]
	5.9 Log_10_ PFU/mL; ABM	Nigerian (DAK 1279)	MIR: 11%, DIR: 0%	[70]
Cairns, Australia	7.2 Log_10_ TCID_50_/mL; ABM	Nigerian (BA-55)	IR: 80%, DIR: 72%, ^9^ TIR: 52% at 14 DPF	[71]
	6.7 Log_10_ TCID_50_/mL; ABM	Bolivian (Cinetrop 28)	IR: 24%, DIR: 24%, TIR: 24% at 15 DPF	[71]
	8 Log_10_ TCID_50_/mL; ABM	African (Asibi)	IR: 92%, DIR: 80% At 14 DPF	[71]
Townsville, Australia	7.2 Log_10_ TCID_50_/mL; ABM	Nigerian (BA-55)	IR: 72%, DIR: 60%, TIR: 60% at 14 DPF	[71]
	6.7 Log_10_ TCID_50_/mL; ABM	Bolivian (Cinetrop 28)	IR: 36%, DIR: 32%, TIR: 28% at 14 DPF	[71]
	8 Log_10_ TCID_50_/mL ABM	African (Asibi)	IR: 96%, DIR: 100%	[71]
Cambodia, Asia	7 Log_10_ PFU/mL; ABM	Senegalese (S-79)	IR: 40%, DR: 60%, TR: 0% at 14 DPF IR: 40%, DR: 90%, TR: 35% at 21 DPF	[73]
Vietnam, Asia	7 Log_10_ PFU/mL; ABM	Senegalese (S-79)	IR: 60%, DR: 70%, TR: 0% at 14 DPF IR: 70%, DR: 55%, TR: 5% at 21 DPF	[73]
Trung, Asia	7 Log_10_ PFU/mL; ABM	Senegalese (S-79)	IR: 95%, DR: 75%, TR: 0% at 14 DPF IR: 90%, DR: 90%, TR: 30% at 21 DPF	[73]
Laos, Asia	7 Log_10_ PFU/mL; ABM	Senegalese (S-79)	IR: 75%, DR: 62%, TR: 0% at 14 DPF IR: 100%, DR: 90%, TR: 40% at 21 DPF	[73]
Thailand, Asia	7 Log_10_ PFU/mL; ABM	Senegalese (S-79)	IR: 100%, DR: 85%, TR: 45% at 14 DPF IR: 100%, DR: 100%, TR: 58% at 21 DPF	[73]
Singapore, Asia	7 Log_10_ PFU/mL; ABM	Senegalese (S-79)	IR: 80%, DR: 80%, TR: 10% at 14 DPF IR: 100%, DR: 50%, TR: 5% at 21 DPF	[73]
New Caledonia, Asia	7 Log_10_ PFU/mL; ABM	Senegalese (S-79)	IR: 100%, DR: 100%, TR: 10% at 14 DPF IR: 100%, DR: 75%, TR: 25% at 21 DPF	[73]
Taiwan, Asia	7 Log_10_ PFU/mL; ABM	Senegalese (S-79)	IR: 65%, DR: 55%, TR: 0% at 14 DPF IR: 85%, DR: 50%, TR: 50% at 21 DPF	[73]

^1^ PFU, plaque-forming units; ^2^ ABM, artificial bloodmeal; ^3^ DPF, days post-feeding; ^4^ IR, infection rate (number of positive mosquitoes/total mosquitoes); ^5^ DR, dissemination rate (number of positive heads or legs/total infected mosquitoes); ^6^ TR, transmission rate (number of positive saliva or salivary glands/total infected mosquitoes); ^7^ MIR, midgut infection rate (number of positive midguts/total engorged mosquitoes); ^8^ DIR, disseminated infection rate (number of positive legs/total engorged mosquitoes); ^9^ TIR, transmitted infection rate (number of positive saliva or salivary glands/total engorged mosquitoes).

### 4.2. The Sylvatic Cycle of Transmission

YFV is maintained in the sylvatic/enzootic cycle in South America and sub-Saharan Africa between non-human primates and arboreal mosquitoes. The principal sylvatic vector in some regions of tropical Africa is *Ae. africanus*; however, other mosquito species such as *Ae. furcifer- taylori*, *Ae. vittatus*, *Ae. simpsoni*, etc., have also been implicated [3]. These mosquitoes are primatophilic and usually use treeholes filled with rainwater as oviposition sites and larval habitats [74]. The occurrence of YFV in East Africa has solely been attributed to sylvatic vectors [75]. Unlike Africa, the sylvatic vectors in South America belong to the genera *Haemogogus* spp. (*Hg. janthinomys*, *Hg. spegazzinni*, *Hg. leucocelaenus*) and *Sabethes* spp. (*Sa. chloropterus*, *Sa. albipivus*, *Sa. soperi*, *Sa. cyaneus*) [3]. *Haemogogus leucocelaenus* and *Sa. albipivirus* experimentally infected using artificial bloodmeals at a titer of 10^6^ PFU/mL were competent for West African and Brazilian strains of YFV with approximately 48% infection [66].

YFV has been isolated from various sylvatic mosquito vectors in East and West Africa and South America [5,76,77]. However, these sylvatic mosquitoes are generally difficult to rear in the laboratory; hence, the knowledge of their biology is often restricted to the study of field-collected mosquitoes. Unfortunately, this limits our understanding of vector competence, transmission, and susceptibility of these enzootic vectors.

Eighty-nine species of mosquitoes collected from 44 municipalities of five Brazilian states during the 2016–2019 YF outbreak revealed that five sylvatic species were positive in 42% of municipalities [6]. *Haemogogus janthinomys* and *Hg. leucocelaenus* were found to be the primary vectors due to high numbers of YFV-positive pools and the extensive distribution of the mosquito population [6]. Despite the wide distribution of the urban vector, *Ae. aegypti*, YFV was not detected in the mosquito pools of this species.

## 5. Discussion and Conclusions

There have been excellent advances in our understanding of YF; however, there is much about the virus that we do not know. Fatal outcome in YFV-infected patients has been associated with elevated cytokines and chemokines such as interleukin (IL)-6, monocyte chemoattractant protein (MCP)-1, interferon-inducible protein (IP)-10, tumor necrosis factor (TNF)-α, and IL1 receptor antagonist (IL1-RA) [78]. Severe liver damage accompanied by elevated levels of ALT and AST, coagulopathy, kidney failure, and rapid deterioration has been observed in severe clinical cases [75]. Histopathology studies of YF-infected patient livers have demonstrated high prevalence of apoptosis over necrosis, steatosis, and councilman bodies corresponding to hepatocyte damage from apoptosis [79,80]. Most of these observations have been studied in all three animal models showing characteristic viscerotropic disease and molecular abnormalities, especially in the NHP model. Unfortunately, the NHP model being the most relevant to represent human disease is also more difficult, expensive, and ethically challenging to obtain. Mice and hamsters are easily available; however, they each have their restrictions with respect to using immunodeficient mice or hamster-adapted strains.

There is great variability in the susceptibility of *Ae. aegypti* mosquitoes to YFV infection across geographic vector populations as well as viral genotypes, strains, and dose [66,67,68,69,70,81]. Despite numerous efforts, it is difficult to standardize vector competence studies due to differences in laboratory conditions, mosquito rearing, food and water, microbiome, insectary environment, and colonization history [82]. Importantly, more studies are required to bridge the gap between results from laboratory studies and data from epidemic emergence. The lack of YFV in the urban vector, *Ae. aegypti*, along with the lack of human cases distant from infected NHPs, have demonstrated that the virus failed to establish itself in the urban, human-amplified cycle during the 2016 Brazilian outbreak. This was surprising considering the detection of NHP and human cases in major cities with little human immunity due to the lack of vaccination during many years without outbreaks. Moreover, the wide presence of competent *Ae. aegypti* throughout Asia [73,83] yet complete absence of YFV is a similar mystery yet to be solved [84]. Multiple theories to explain the lack of YFV have been posited, which include the failure of virus introductions into Asia (although other viruses with similar origins in Africa such as Zika and chikungunya have extensive histories of movement to Asia), competition or interference with other flaviviruses such as dengue within the vector (although vector infection rates are low even during epidemics) and cross-protection provided by other flaviviruses such as dengue, which are hyperendemic in much of Asia and Latin America [84,85]. The two major outbreaks of YFV in Angola and Brazil during the 2015–2019 period as well as the more recent epizootics reported in Brazil [86] underscore the need to reassess the epidemic emergence potential of the virus. Additionally, we need to continue our attempts to develop better animal models for antiviral and vaccine development as well as standardize the vector competence studies across laboratories to minimize variability in results and improve interpretation of results from different research groups.

## Figures and Tables

**Figure 1 microorganisms-10-01578-f001:**
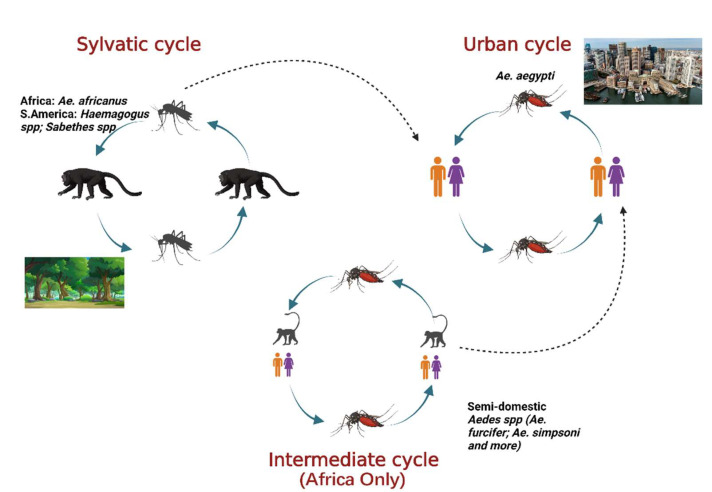
The three cycles for YFV transmission involve various mosquito species and hosts. In the sylvatic cycle, the YFV is maintained between sylvatic mosquito species and NHPs as host. In the urban cycle, the virus is primarily maintained between *Ae. aegypti* mosquitoes and humans as host. The intermediate cycle occurs in Africa only in moist savannah regions with small human settlements. The virus can be transmitted from semi-domestic mosquitoes to humans or NHPs as host. Adapted from CDC (created with Biorender.com, accessed on 1 July 2022).

**Figure 2 microorganisms-10-01578-f002:**
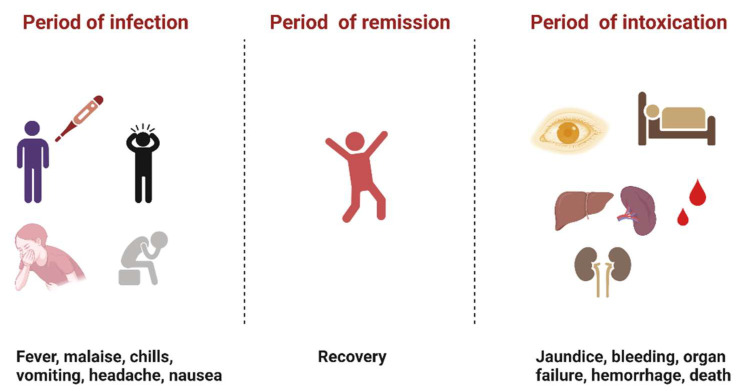
The three phases of severe yellow fever disease: period of infection characterized by non-specific symptoms, period of remission, where most individuals recover, and period of intoxication that occurs in extreme cases where individuals progress to a more severe form of the disease [12]. (Created with Biorender.com, accessed on 1 July 2022).

**Table 3 microorganisms-10-01578-t003:** Advantages and disadvantage of animal models of YF.

Animal model	Advantages	Disadvantages
Mice	Small, easy to handle, cost-effective; reagents available; can conduct studies involving large number of animals; can use wild-type strains of YFV	Need to use immunocompromised mice; generally, viremia is not high enough to conduct mosquito experiments; disease outcome not similar to humans
Hamster	Viscerotropic disease outcome typical of YFV; can use immune-intact animals; generates high enough viremia for mosquito experiments	Need to use hamster-adapted strains; reagents are limited
Non-human primates	Disease outcome similar to humans; intact immunity; best representative model for vaccine and therapeutic studies; generally required in the past prior to clinical trials	Expensive; require specialized facilities, training; cannot use large groups due to ethical restrictions

## Data Availability

Not applicable.

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
