# Peer review of "Yellow Fever: Roles of Animal Models and Arthropod Vector Studies in Understanding Epidemic Emergence"

_microorganisms, 2022, doi:10.3390/microorganisms10081578_

Round 1

Reviewer 1 Report

This is a well-written, informative and enjoyable short review on animal models of yellow fever by Shinde et al.,. I only have a few minor comments that may be of use here. 

Reference for taxonomy of Flaviviridae - JGV: https://www.microbiologyresearch.org/content/journal/jgv/10.1099/jgv.0.000672

Mention that it is a live vaccine and so relatively hard to make in bulk so doses not highly available and not everyone can take it due to the chance it might cause disease (e.g. in older individuals).

Make clear whether the outlined phases of infection are from a reference or are constructed by the author.

Line 65 - change to YFV

Line 77 - define YF as yellow fever

In table, for each strain, in what cell line is infectivity determined in pfu and tcid50?

Italicise all latin binomial names

Add in mouse as mus musculus

Line 184 : make sure it is 25 to 5 x 10^4

Final section reads more like "outlook and future directions" - a better concluding statement possibly focusing on the pros and cons of each model in light of human disease and interventions (like in table 3) would be useful.

Has there been no attempts to infect hSTAT2 knock in mice with YFV?

Just a comment about the general accuracy of animal models but how do we know the models are like the human disease at a molecular level? What is know about the human pathogenesis at a molecular level in terms of genes/proteins/pathways changing in liver or blood?

Reviewer 2 Report

Introduction

Line 36+ - Please give a little more detail on the intermediate cycle. Such as how humans get infected. Do they get bitten by infected sylvatic mosquitoes when they go into the forest or have these mosquitos changed their host preference to include humans and enter houses?

Could YFV infection not occur in rural/contact zones in South America as well or does it only occur frequently enough in Africa to warrant a name? The citation for this information is only listed as 03774.pdf. The authors and journal should be indicated as they are included in the pdf.

Line 42 – YF has not been defined.

Line 52 – “ranging” has already been used in this sentence.

Figure 2 – Does not provide any additional information.

Line 70 – “…as well as…” The position of this phrase makes this sentence awkward. The phrase describes Dr. Finlay and should be closer to its subject.

Line 99 – WT has not been defined.

Line 105 – What does Three-4 mean?

Line 113 – What are IFNAR-/- mice? What are they missing? C57/BL6 is different from that on

Line 120 and in Table 1.

Line 119-120 – Please describe FRG mice in more detail. How do the

Table 1: Abbreviations need to be described in footnotes.

Line 133 – Mesocricetus auratus should be italicized.

Line 149 – DPI has not been defined.

Line 153 – can “be” utilized

Line 160 – Why are page numbers included?

Line 198+ - Ensure that all binomial and genus names are italicized.

Line 201 – This study is not included in Table2

Table 3 should be discussed. There is no reference to it in the text.

Line 215 – What other animals have been attempted?

Section 4 – This section would benefit from inclusion of YFV surveillance data and any transmission data from these studies (ability to become infected and transmit are important).

Also to match the other sections a table of surveillance and/or vector competence studies should be added.

For urban cycle section, if Aedes aegypti aegypti are the predominant vector then the information discussed should focus on this subspecies. Adding information on other Aedes aegypti subspecies is confusing for the reader.

The transmission cycles were addressed previously and do not need to discussed in detail in this section.

The discussion of vector control and elimination of urban outbreaks should be included in the history section or in the discussion, rather than here.

Lines 249-254 – If the 2016-2019 Brazil outbreak was from sylvatic mosquitoes why is it mentioned in the urban transmission section? Were the Aedes aegypti mosquitoes found in the areas affected by the outbreak known to be competent for YFV? How close were the outbreak areas to urban centers that there was concern that YFV could enter the urban areas?

Line 255-260 – These sentences do not lead in to the data well. They would be more appropriate for the discussion.

Line 261 – There is no way for the reader to know that this data refers to the statement made on Line 254.  

Line 267 – Midgut infection rate and dissemination rate do not need to be capitalized.

The Discussion/Conclusions section needs to tie the information presented in the different sections back to the point of the review, which was described as the roles of animal models and arthropod vector studies in understanding epidemic emergence. The data presented needs to be integrated into the bigger picture and the authors should discuss gaps that need to be addressed in order to understand emergence.

References

The citations are inconsistent with some titles having the first word capitalized and others with all words capitalized, some journal names are written out while others are abbreviated. Most concerning are the references to papers accessed only that do not list the authors, title, date and journal, for example #4.
